# Association of Work Patterns and Periodontitis Prevalence in Korean Adults Aged 50 Years or Older: A Nationwide Representative Study

**DOI:** 10.3390/ijerph17114006

**Published:** 2020-06-04

**Authors:** Young Jin Ra, Young Jin Tak, Yun Jin Kim, Sang Yeoup Lee, Jeong Gyu Lee, Yu Hyeon Yi, Young Hye Cho, Hye Rim Hwang, Seung Hun Lee, Eun Ju Park, Young In Lee

**Affiliations:** 1Department of Family Medicine, Pusan National University School of Medicine, Yangsan 50612, Korea; yjra@pnuh.co.kr (Y.J.R.); yujkim@pusan.ac.kr (Y.J.K.); eltidine@hanmail.net (J.G.L.); eeugus@hanmail.net (Y.H.Y.); hezera83@naver.com (H.R.H.); b8u5ey@hanmail.net (S.H.L.); 2Biomedical Research Institute, Pusan National University Hospital, Busan 49241, Korea; 3Family Medicine Clinic, Obesity, Metabolism and Nutrition Center and Research Institute of Convergence of Biomedical Science and Technology, Pusan National University Yangsan Hospital, Yangsan 50612, Korea; saylee@pusan.ac.kr (S.Y.L.); younghye82@naver.com (Y.H.C.); everblue124@daum.net (E.J.P.); ylee23@gmail.com (Y.I.L.); 4Department of Medical Education, Pusan National University School of Medicine, Yangsan 50612, Korea; 5Busan Tobacco Control Center, Pusan National University Hospital, Busan 49241, Korea

**Keywords:** work pattern, periodontitis, oral health care, regular oral examinations

## Abstract

This study analyzed the relationship between the work pattern and the prevalence of periodontitis. We analyzed the data of 3320 adults (1779 men, 1543 women) aged 51–80 years from the Korean National Health and Nutrition Examination Survey (2013–2015). The work pattern was divided into two groups (regular and irregular). The periodontal status was assessed using the community periodontal index. We observed a statistically significant difference in the association between work patterns and prevalence of periodontitis in Korean women aged over 50 years. For female workers with irregular work patterns, the prevalence of periodontitis was lower than that in workers with regular work patterns by 10.3% (40.3% vs. 30.0%, *p* = 0.011). The annual health examination rate was significantly higher in the irregular group than in the regular group (for men 77.9% vs. 73.5%; *p* < 0.001, for women 76.4% vs. 75.9%; *p* < 0.001). In female workers with irregular work patterns, the annual dental examination rate was significantly higher than that in workers with a regular work pattern by 7.7% (34.3% vs. 26.6%, *p* = 0.043). In conclusion we found a statistically significant difference between the work patterns and prevalence of periodontitis in Korean women aged over 50 years.

## 1. Introduction

Increasing economic growth and developments in science and technology have led to improvements in quality of life and human life expectancy. Oral health has gained considerable attention as it plays an important role in maintaining overall health [1]. According to the statistical data published by the Health Insurance Review and Assessment Service, the second major outpatient disease is periodontitis while the sixth is tooth decay [2]. Periodontitis is a chronic inflammatory condition where a reaction between the dental plaque and host tissue destroys the periodontal tissue. There are three identified forms of periodontitis: necrotizing periodontitis, periodontitis as a manifestation of systemic disease, and the forms of the disease previously recognized “chronic” or “aggressive” [3]. Periodontitis has become increasingly prevalent and is reported to be a major cause of tooth loss in adults over 35 years of age [4]. The risk factors for periodontitis include bacterial infection, older age, low socioeconomic status, psychiatric condition, family history, poor oral hygiene, smoking, and certain systemic diseases [4]. These risk factors affect the immune response and increase the risk of periodontitis [5], therefore, they should be considered for its prevention and treatment [6]. Periodontitis is mainly the result of infection and inflammation of the gum and bone tissue that surrounds and supports the teeth. Periodontitis and tooth decay are the two major threats to dental health [7]. Periodontitis is caused a combination of risk factors; gingivitis, gingival hemorrhage and edema, gingival recession, periodontal papilla formation, alveolar bone absorption, and tooth irregularity arising from a bacterial infection in the periphery of the tooth and the periodontal pocket, all cause damage to the surrounding tissue resulting in tooth loss. Oral bacteria can infect the tissue surrounding the tooth, causing inflammation that leads to periodontal disease. When bacteria are present on the teeth for a long period, they form a film called plaque, which eventually hardens to become tartar (also known as calculus). The build-up of tartar can spread below the gum line, making the teeth harder to clean; by this stage, only a dental health professional can remove the tartar and stop the periodontal disease progression [7]. In early periodontitis, there are no subjective symptoms; however, as the disease becomes chronic, it leads to tooth loss, poor chewing ability, inadequate nutrition, and poor quality of life [8]. The immediate local causes of periodontitis are, therefore, biofilm, calculus, and interrelated trauma. However, there is also a range of systemic causes, including malnutrition, endocrine disorders, use of certain drugs, smoking, and stress [9]. Effective preventative strategies include brushing the teeth to manage dental hygiene, using oral hygiene products, and regular oral examinations and scaling [10]. Oral health behaviors have been reported to influence the risk of developing periodontitis [11,12]. Recent studies suggest links between periodontitis and smoking, diabetes, obesity, and chronic obstructive pulmonary disease [13,14,15,16]. It has also been suggested that periodontitis can increase the risk of coronary artery disease, stroke, preterm birth, low birth weight, and respiratory disease [17].

The Korean survey of working conditions reported that the proportion of employees involved in shift work has increased from 7.2% in 2006 to 10.9% in 2010 [18]. In that survey, the most commonly reported type of shift work was two regular shifts (38.6%), followed by three regular shifts (23.9%), 24-h rotating shifts (14.0%), two irregular shifts (5.7%), fixed shifts (5.5%), split shifts on weekdays (3.6%), three irregular shifts (2.9%), and other shifts (5.8%) [18]. In modern society, industrialization and changes in the social environment have led to an increase in the number of industries with uninterrupted production, such as producers of petrochemicals, semiconductors, iron, and steel. The number of shifts that can be worked has increased as more and more production facilities are operating 24 h a day, 7 days a week, to recover their investment costs. In this paper, irregular work pattern refers to work that occurs outside the 8-h working day of 9 a.m. to 5 p.m., and may include early morning, late afternoon, or night shifts. In the USA, 20%–25% of workers are reportedly engaged in rotating shifts, evening, or night shift work [19]. According to a study performed in Korea, about 2% of the manufacturing-related companies operate a shift work system [20]. There has been increasing research on the effects of shift work on workers’ health in other countries. In the short term, workers on rotating shifts and those who work at night have suboptimal health due to lack of regular sleeping and eating patterns. In the long term, these workers have increased incidence rates of obesity, diabetes, metabolic syndrome, cardiovascular disease, gastrointestinal disease, and cancer [21,22,23,24]. Night shift workers are more likely to experience sleep deprivation due to their irregular sleeping times and disruption of circadian rhythms, which may contribute to the development of stress-related mental problems in vulnerable individuals. Furthermore, workers with stress-related neuropsychiatric disorders have been found to have changes in glucocorticoid-related functional connectivity [25]. Therefore, shift work is likely to have a negative impact on mental health [26]. It is well known that shift work is associated with an increased risk of depression. One study in Korea found that shift work was independently associated with suicidal ideation in men [27]. The pathway via which shift work results in higher suicidality is likely related to depression. Many studies about shift workers have identified irregular sleep and lack of sleep as important sources of health problems. Sleep deprivation causes drowsiness, fatigue, anxiety, tension, and loss of concentration, all of which affect physical functioning during the day [28]. Moreover, chronic sleep problems affect the immune response. Therefore, good quality sleep is essential for the promotion and maintenance of health [29]. Many studies have shown that irregular work pattern can lead to cardiovascular disease [30], sleep disorders [31], peptic ulcer [22], and breast cancer [32]. One study found that working night shift causes changes in thyroid-stimulating hormone levels and suggested that night shift workers have an increased risk of thyroid disease [33]. 

The purpose of this study was to investigate whether there were differences in the prevalence of periodontitis between work patterns. In addition, we were interested in how important the “work patterns” plays in the difference in prevalence of periodontitis. 

## 2. Materials and Methods 

### 2.1. Study Population

This representative, cross-sectional, nationwide study used data from the Korea National Health and Nutrition Examination Survey (KNHANES) VI (2013–2015), performed by the Korea Centers for Disease Control and Prevention. The KNHANES consists of three components, i.e., a health interview, a health examination, and a nutrition survey. Its goal was to assess the health and nutritional status of the Korean population. A nationally representative sample was selected from the Korean population using household records developed by the 2005 Population and Housing Census in Korea. Twenty households from each district were selected using a stratified, multistage probability cluster sampling system that considers the age, sex, and geographic location of each of the study participants. In total, 22,948 individuals from 9491 households were interviewed in the survey. The present study includes data for elderly individuals (aged ≥50 years; Figure 1) with data available for periodontitis status and responses to the work pattern questionnaire. We excluded respondents who answered “no” to the question about whether or not they were currently working, those who were unable to distinguish between different types of work, and those for whom baseline data were incomplete. After the application of the exclusion criteria, data for 3320 respondents were available for analysis. The KNHANES was approved by the Institutional Review Board of the Korean Centers for Disease Control and Prevention. (IRB No. H-1907-029-081). All study participants provided informed consent. 

### 2.2. General Characteristics of the Study Participants

The KNHANES collected data on participants’ socio-demographics, lifestyle behavior, and medical and family history using questionnaires and anthropometric measurements. In this study, each respondent’s monthly income was categorized into quartiles of low, lower-middle, upper-middle, or high. Educational status was categorized as graduated from elementary school (or lower), middle school, high school, or college or higher. Smoking status was self-reported and classified according to the Korean Health and Nutrition Examination Survey. Men who consumed ≥7 standard drinks and women who consumed ≥5 standard drinks of alcohol at least twice per week were classified as high-risk drinkers. Body mass index (BMI) was calculated as weight in kilograms divided by the square of height in meters (kg/m^2^). The Asian obesity standard was used in this study, whereby underweight was defined as a BMI <18.5, normal as a BMI of 18.5–22.9, overweight as a BMI of 23–24.9, and obesity as a BMI >25. The presence of common chronic diseases, i.e., hypertension, diabetes mellitus, and dyslipidemia, was also evaluated.

### 2.3. Work Pattern

According to their responses to the questionnaire, the study participants were categorized according to the regular or irregular work pattern group. Regular work pattern was defined as work performed the normal 8-h day (9:00 a.m. to 5:00 p.m.). Irregular work pattern was defined as work performed outside the normal 8-h day (9:00 a.m. to 5:00 p.m.) and included early morning, late afternoon, night work including any kind of shift work.

### 2.4. Oral Examination

All subjects underwent a dental examination by a dental professional, who evaluated the number of remaining teeth and the presence of dental prosthetics, dental implants, and periodontal disease, as well as a need for further dental procedures. Self-reported oral care-related behaviors were noted, such as level of oral health awareness, tooth brushing, use of oral care products, and attendance for annual dental examinations as well as the presence of any untreated lesions in the oral cavity. Tooth brushing was defined as brushing the teeth at least three times a day. The universal numbering system developed by the Fédération Dentaire Internationale was used to provide information on a specific tooth. 

### 2.5. Periodontal Status

Periodontal tissue status was assessed using the community periodontal index (CPI) scoring system (0—healthy; 1—bleeding; 2—presence of calculus; 3—periodontal pocket 4–5 mm; 4—periodontal pocket ≥6 mm). The maximum CPI score for each reference tooth at the six sites determined each subject’s overall score. According to the final CPI score, each subject was classified as having a healthy periodontal condition (0–2) or periodontal disease (3–4).

### 2.6. Statistical Analysis

The KNHANES data for 2013, 2014, and 2015 were combined. The sampling results were weighted to allow for nationally representative prevalence estimates in the Korean population. The weights were calculated by accounting for the complex survey design, nonresponse rate, and poststratification. The study participants were assumed to represent the Korean population after the weighting of the data. The clinical characteristics of the study population were compared based on gender using Pearson’s chi-squared test for categorical variables and a generalized linear model for continuous variables. The association between work patterns and periodontitis was investigated using Pearson’s chi-squared test. A multivariate-adjusted logistic regression analysis that included three models was used to evaluate the relationship between work patterns and periodontitis. The first model was adjusted for age, the second for age with the addition of prevalence of diabetes mellitus, alcohol consumption, smoking status, and annual health checkup status, and the third for the same variables with the addition of annual dental examination status. The data are shown as the mean and standard error or as the frequency (percentage). All statistical analyses were performed using SPSS for Windows (version 23.0; IBM Corp., Armonk, NY, USA). A p-value <0.05 was considered to be statistically significant. 

## 3. Results

### 3.1. Characteristics of the Study Population at Baseline

In total, 3320 participants were included in the study. Characteristics of the study population at baseline are shown in Table 1. The study population included 1779 men (53.6%) and 1543 women (46.4%) with a mean age of 60.86 ± 7.32 years. In the characteristics of the study population at baseline, we compared and analyzed family income, education level, smoking, alcohol, obesity, hypertension, diabetes and dyslipidemia.

There was a statistically significant difference in the educational level between the two study groups. (*p* < 0.001). In the irregular work pattern group, the percentage of “graduation from elementary school or lower” level “was highest. (37.1%). On the other hand, in the regular work pattern group, the percentage of “high school graduate” level was highest (36.9%). There was a statistically significant difference in the smoking status and the proportion of patients with dyslipidemia between the two study groups. The proportion of smokers in the irregular work pattern group was higher than in the regular work pattern group (17.2% vs. 21.3; *p* = 0.014). The proportion of patients with dyslipidemia was high in the irregular work pattern group (19.9% vs. 23.3%; *p* = 0.041).

There were no statistically significant differences in family income, alcohol, obesity including BMI, hypertension, and diabetes.

### 3.2. Health Behaviors and Lifestyle-Related Disorders

Health behaviors and lifestyle-related disorders (diabetes, hypertension, hyperlipidemia, smoking, drinking, sleeping time, stress level) were also compared between men and women. The annual health examination rate was significantly higher in the irregular work pattern group than that in the regular work pattern group for men (77.9% vs. 73.5%; *p* < 0.001) and women (76.4% vs. 75.9%; *p* < 0.001). The proportion of women who smoked was significantly higher in the irregular work pattern group (6.8% vs. 2.8%; *p* = 0.013).

There was no statistically significant between-group difference in alcohol consumption, sleeping time, or stress levels (Table 2).

### 3.3. Dental Health Care Behaviors

Oral health behaviors (tooth brushing, use of oral hygiene products, tooth decay, annual dental/oral examination, attendance at a dental clinic in the previous year) were also investigated according to the gender and work pattern. The annual dental examination rate was significantly higher among women in the irregular work pattern group than that in the regular work pattern group (34.3% vs. 26.6%; *p* = 0.043). There were no statistically significant between-group gender-related differences in other dental health care behaviors (Table 3).

### 3.4. Association between Work Pattern and Periodontitis

A statistically significant difference was observed between the type of work pattern and the prevalence of periodontitis in women. Women with an irregular work pattern had a lower prevalence of periodontitis than women with a regular work pattern. For 1543 women over 50 years of age, the prevalence of periodontitis was significantly higher in the regular work pattern group than that in the irregular work pattern group (40.3% vs. 30.0%; *p* = 0.011). 

However, there was no significant difference in the prevalence of periodontitis between men in the regular and irregular work pattern groups (55.1% vs. 56.4%, *p* = 0.742; Figure 2).

In the model adjusted for all confounding factors, there was a significant difference in the relationship between the work pattern and prevalence of periodontitis in women (Table 4).

## 4. Discussion

This study investigated the relationship between work patterns (working regular hours or working shifts) and the prevalence of periodontitis in workers aged 50 years or older using data from a large sample of the general population in Korea. The reason for selecting this age group is because the statistical data in Korea indicate an increase in the prevalence of periodontitis from the age of 40 s years onwards and a persistently high level from the age of 50 s years onwards.

Women with an irregular work pattern had a lower prevalence of periodontitis than those with a regular work pattern. For 1543 women over 50 years of age, the prevalence of periodontitis was statistically significantly higher in the regular work pattern group than in the irregular work pattern group (40.3% vs. 30.0%; *p* = 0.011; Figure 2). In contrast, there was no statistically significant difference in the prevalence of periodontitis in 1779 men over 50 years of age according to whether their work pattern was regular or irregular (55.1% vs. 56.40%; *p* = 0.742; Figure 2). In the model adjusted for all confounding factors, there was a statistically significant relationship between the work pattern and prevalence of periodontitis only in women (Table 4). 

We analyzed the general health behavior factors that are thought to be related to the prevalence of periodontitis. Smoking, hyperlipidemia, and low educational attainment were significantly more common in groups with an irregular working pattern (Table 1). However, rates of attendance for an annual health examination was significantly higher among workers with an irregular work pattern, particularly in women. However, there was no statistically significant difference between the groups for alcohol consumption, sleep time, or stress levels (Table 2). Investigation of dental health care behaviors confirmed that the annual dental examination rate of female workers in the irregular working pattern group was higher than the annual dental examination rate of female workers in the regular working pattern group, which was a statistically significant difference (34.3% vs. 26.6%; *p* = 0.043) (Table 3).

In this paper, the prevalence of periodontitis according to the work pattern differed between men and women over 50. Although not statistically significant, the prevalence of periodontitis in men was somewhat lower in the regular working pattern group. (55.1% vs. 56.4%, *p* = 0.742; Figure 2). 

For men, it is thought that lifestyle behaviors that are harmful to periodontitis such as smoking and alcohol consumption have a more complex effect on the prevalence of periodontal disease than women. Therefore, it was difficult to clearly explain the relationship between the prevalence of periodontal disease according to the working pattern in men. 

For women, there was a significant difference in the effect of oral examinations on the prevalence of periodontitis between those who worked with regular work pattern and those who worked with irregular work pattern. Unlike men’s results, the prevalence of periodontitis was lower in women with irregular work pattern groups. In another study of women in their 50 s or older who worked shifts, shift work was associated with unfavorable health behaviors, including an increased risk of smoking (OR (odds ratio) 1.73; 95% CI (confidence interval) 1.34–2.22) and inadequate sleep (OR 1.24; 95% CI 1.05–1.47) when compared to their counterparts who worked during regular daytime hours [34]. Smoking, which is a major risk factor for periodontal disease, was significantly more common in irregular work pattern group, whereas the rate of periodontal disease was lower. Moreover, the effects of irregular sleep times on immune status and hormone balance were expected and these physiological changes, if sustained, may lead to differences in health-related indices in groups working irregular shifts. Given that oral health screening was significantly more common in irregular shift workers, it is probable that oral examinations cannot offset the effects of smoking, alcohol consumption, diabetes, and irregular sleeping times, which are well-known risk factors for periodontal disease. This means that workers with irregular working patterns were given more opportunities to receive oral examinations more frequently than those who had to work during the daytime (9:00 a.m. to 5:00 p.m.) and this contributes to the prevalence of periodontitis for female workers. Regular oral examinations are expected to significantly contribute to a reduction of the prevalence of periodontitis among shift workers. Therefore, to reduce the prevalence of periodontitis, an increased institutional effort is required to address dental health issues related to socioeconomic inequality, such as access to regular oral examinations and health screening examinations.

This study had several limitations. First, it had a cross-sectional design, which makes it difficult to assign causality to the relationships identified between work patterns and periodontitis. Second, the data analyzed were self-reported, so the possibility of reporting bias cannot be ruled out. Third, it was not possible to adjust for all potentially confounding factors. However, these shortcomings may be offset by several strengths, including the large-scale of the study and the fact that the data are representative of the general Korean population. Its complex design not only ensured representativeness but also the reliability of the data. Second, we used a validated tool (the CPI score) as an indicator of the prevalence of periodontal disease. Moreover, we considered and adjusted for more variables related to work patterns and the prevalence of periodontitis than any of the other relevant published studies. Third, and most important, is that this is the first paper to explore the link between work patterns and the prevalence of periodontitis.

## 5. Conclusions

In this study, we found a statistically significant difference between the work patterns and prevalence of periodontitis in Korean women aged over 50 years. For female workers with irregular working patterns, the prevalence of periodontitis was lower by 10.3% than that of workers with regular working patterns (40.3% vs. 30.0%, *p* = 0.011). In contrast, there was no statistically significant difference in the prevalence of periodontitis in male workers over 50 years of age according to their work patterns (55.1% vs. 56.40%, *p* = 0.742).

## Figures and Tables

**Figure 1 ijerph-17-04006-f001:**
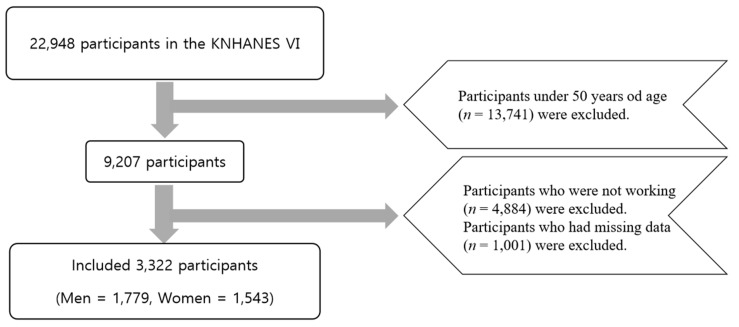
Diagram showing the flow of participants through the study. KNHANES, Korea National Health and Nutrition Examination Survey.

**Figure 2 ijerph-17-04006-f002:**
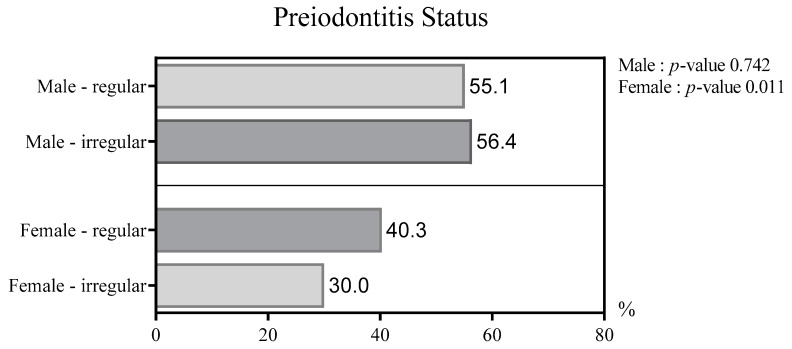
Periodontitis status according to gender and work pattern.

**Table 1 ijerph-17-04006-t001:** Characteristics of the study population at baseline.

		Total (*n* = 3320)	Work Pattern		Weighted		*p*-Value
			Regular	Irregular	Regular	Irregular	
Sex	Male	1779 (53.6%)	1504 (53.3%)	275 (55.2%)	58.6%	59.5%	0.756
	Female	1543 (46.4%)	1320 (46.7%)	223 (44.8%)	41.4%	40.5%	
Age, years		60.86 ± 7.32	60.99 ± 7.40	60.12 ± 6.84	59.40 ± 0.19	58.83 ± 0.34	0.133
Family income, percentile, %	25th	629 (18.9%)	565 (20.0%)	64 (12.9%)	16.9%	12.5%	0.065
	25–50th	957 (28.8%)	808 (28.6%)	149 (29.9%)	26.8%	27.6%	
	50–75th	810 (24.4%)	663 (23.5%)	147 (29.5%)	25.4%	30.9%	
	75–100th	926 (27.9%)	788 (27.9%)	138 (27.7%)	30.9%	29.0%	
Education level, %	Elementary school or less	1196 (36.0%)	1048 (37.1%)	148 (29.7%)	32.4%	25.9%	<0.001
	Middle school graduate	664 (20.0%)	567 (20.1%)	97 (19.5%)	20.6%	18.4%	
	High school graduate	894 (26.9%)	710 (25.1%)	184 (36.9%)	27.0%	39.4%	
	College graduate or higher	568 (17.1%)	499 (17.7%)	69 (13.9%)	20.1%	16.3%	
Smoking status, %	Yes	593 (17.9%)	487 (17.2%)	106 (21.3%)	19.1%	24.9%	0.014
Alcohol consumption, %	High risk	461 (13.9%)	402 (14.2%)	59 (11.8%)	17.0%	14.2%	0.248
Obesity	BMI >25	1249 (37.6%)	1066 (37.7%)	183 (36.7%)	38.0%	36.6%	0.867
BMI		24.26 ± 3.01	24.28 ± 3.02	24.13 ± 2.95	24.29 ± 0.07	24.13 ± 0.14	0.307
HTN	Yes	1084 (32.6%)	937 (33.2%)	147 (29.5%)	29.8%	27.9%	0.463
DM	Yes	382 (11.5%)	318 (11.3%)	64 (12.9%)	10.6%	13.5%	0.114
Dyslipidemia	Yes	678 (20.4%)	562 (19.9%)	116 (23.3%)	18.7%	23.1%	0.041

BMI—body mass index; DM—diabetes mellitus; HTN—hypertension.

**Table 2 ijerph-17-04006-t002:** Health behaviors and lifestyle conditions according to gender and work pattern.

		Male Workers		*p*-Value	Female Workers		*p*-Value
		Regular Shifts	Irregular Shifts		Regular Shifts	Irregular Shifts	
Diabetes mellitus	Yes	12.4%	15.3%	0.273	8.0%	10.8%	0.264
Hypertension	Yes	8.8%	30.0%	0.73	31.4%	24.9%	0.087
Dyslipidemia	Yes	15.5%	19.5%	0.168	23.2%	28.4%	0.159
Annual health examination	Yes	73.5%	77.9%	<0.001	75.9%	76.4%	<0.001
Smoking status, %	Yes	30.6%	37.3%	0.071	2.8%	6.8%	0.013
Alcohol consumption, %	No	35.5%	38.5%	0.429	78.2%	72.5%	0.112
	Low risk	39.5%	40.9%		16.0%	22.7%	
	High risk	25.0%	20.5%		5.7%	4.8%	
Sleep duration (h/day)		6.71 ± 0.04	6.63 ± 0.09	0.358	6.49 ± 0.04	6.32 ± 0.09	0.09
Stress level	High	2.9%	2.7%	0.977	4.6%	5.0%	0.641
	Moderate	13.6%	13.6%		20.7%	22.2%	

**Table 3 ijerph-17-04006-t003:** Dental health care behaviors according to gender and work pattern.

		Male Workers		*p*-Value	Female Workers		*p*-Value
		Regular Shifts	Irregular Shifts		Regular Shifts	Irregular Shifts	
Dental brushing (%)	No	1.4%	1.1%	0.66	1.5%	0.4%	0.167
	Yes	98.6%	98.9%		98.5%	99.6%	
Use of dental care products	No	58.6%	54.5%	0.325	53.0%	45.7%	0.083
	Yes	41.4%	45.5%		47.0%	54.3%	
Tooth damage	No	81.5%	80.2%	0.674	92.6%	90.1%	0.279
	Yes	18.5%	19.8%		7.4%	9.9%	
Annual dental examination	No	65.1%	66.4%	0.731	73.4%	65.7%	0.043
	Yes	34.9%	33.6%		26.6%	34.3%	
Attendance at a dental clinic	No	43.7%	37.7%	0.122	50.3%	48.6%	0.67
	Yes	56.3%	62.3%		49.7%	51.4%	

**Table 4 ijerph-17-04006-t004:** Odds ratios (95% CI) of the prevalence of periodontitis according to work pattern.

	Model 1		Model 2		Model 3		Model 4	
	OR (95% CI)	*p*-Value	OR (95% CI)	*p*-Value	OR (95% CI)	*p*-Value	OR (95% CI)	*p*-Value
Total	1.14(0.91–1.43)	0.256	1.13 (0.90–1.42)	0.006	1.19 (0.94–1.50)	0.152	1.19 (0.94–1.51)	0.152
Male	0.95 (0.69–1.30)	0.742	0.95 (0.69–1.30)	0.813	0.98 (0.71–1.35)	0.9	0.98 (0.71–1.35)	0.893
Female	1.58 (1.11–2.24)	0.011	1.48 (1.05–2.10)	0	1.57 (1.10–2.24)	0.012	1.56 (1.10–2.22)	0.014

CI—confidence interval; OR—odds ratio. Model 1 was unadjusted; Model 2 was adjusted for age; Model 3 was adjusted for age, the prevalence of diabetes mellitus, alcohol consumption, smoking status, and attendance for annual health examination; and Model 4 was adjusted for the same factors with the addition of attendance for an annual dental examination.

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
