# Peer review of "Association of Work Patterns and Periodontitis Prevalence in Korean Adults Aged 50 Years or Older: A Nationwide Representative Study"

_ijerph, 2020, doi:10.3390/ijerph17114006_

Round 1

Reviewer 1 Report

Dear Authors,

The present study, entitled “Association of work patterns and periodontitis prevalence in Korean adults aged 50 years or older: A nationwide representative study”, needs several modifications, starting by:

The manuscript needs to be sent for an editing service.

Introduction:

The authors are writing about periodontal disease, but they did cite our used the new classification scheme for periodontal disease and conditions. You should read the article below and rewrite the second paragraph.

Caton JG, Armitage G, Berglundh T, Chapple ILC, Jepsen S, Kornman KS, Mealey BL, Papapanou PN, Sanz M, Tonetti MS. A new classification scheme for periodontal and peri-implant diseases and conditions - Introduction and key changes from the 1999 classification. J Clin Periodontol. 2018 Jun;45 Suppl 20:S1-S8. doi: 10.1111/jcpe.12935.

Materials and methods:

The authors should perform in the methods following the STROBE guidelines for observational studies:

von Elm E, Altman DG, Egger M, Pocock SJ, Gøtzsche PC, Vandenbroucke JP; STROBE Initiative. The Strengthening the Reporting of Observational Studies in Epidemiology (STROBE)statement: guidelines for reporting observational studies. J Clin Epidemiol. 2008 Apr;61(4):344-9. PMID: 18313558

Results:

For a scientific paper, the graphics must be prepared in scientific software, such as Prism-GraphPad. It is unacceptable graphics on Microsoft Excel.

Discussion:

The authors should explorer the results in the second and third paragraphs. Because there is no information in both of them.

Conclusion – The authors do not answer the aim of the study in the Conclusion. Why are you using the results of the survey in the Conclusion?

Please, rewrite the Conclusion, and answer the objective of the study.

References – Should be revised in the text.

Regards,

# Reviewer #

Author Response

Response to Reviewer 1 Comments

#Point :

Introduction:

The authors are writing about periodontal disease, but they did cite our used the new classification scheme for periodontal disease and conditions. You should read the article below and rewrite the second paragraph.

Caton JG, Armitage G, Berglundh T, Chapple ILC, Jepsen S, Kornman KS, Mealey BL, Papapanou PN, Sanz M, Tonetti MS. A new classification scheme for periodontal and peri-implant diseases and conditions - Introduction and key changes from the 1999 classification. J Clin Periodontol. 2018 Jun;45 Suppl 20:S1-S8. doi: 10.1111/jcpe.12935.

Response:

Thank you for your keen point and providing reference materials.

The distinct use of "periodontitis" and "periodontal disease" will clarify what this study intends to talk about and may help to avoid unnecessary misunderstandings.

According to the reference material, "periodontal disease" used in this paper has been replaced with "periodontitis". The content of the classification was also added as a reference

p1, line 41

Periodontitis is a chronic inflammatory condition where a reaction between the dental plaque and host tissue destroys the periodontal tissue. There are three forms of periodontitis can be identified necrotizing periodontitis, periodontitis as a manifestation of systemic disease, and the forms of the disease previously recognized “chronic” or “aggressive”. Periodontitis has become increasingly prevalent and is reported to be a major cause of tooth loss in adults over 35 years of age. The risk factors for periodontitis include bacterial infection, older age, low socioeconomic status, psychiatric condition, family history, poor oral hygiene, smoking, and certain systemic diseases . These risk factors affect the immune response and increase the risk of periodontitis, therefore, they should be considered for its prevention and treatment.

#Point :

Materials and methods:

The authors should perform in the methods following the STROBE guidelines for observational studies:

von Elm E, Altman DG, Egger M, Pocock SJ, Gøtzsche PC, Vandenbroucke JP; STROBE Initiative. The Strengthening the Reporting of Observational Studies in Epidemiology (STROBE)statement: guidelines for reporting observational studies. J Clin Epidemiol. 2008 Apr;61(4):344-9. PMID: 18313558

Response: Thank you for pointing out.

Since this paper is a cross-sectional study for large-scale national projects, it is believed that it is independent of the guidelines mentioned.

If there is anything I misunderstand, please point out again.

#Point :

Results:

For a scientific paper, the graphics must be prepared in scientific software, such as Prism-GraphPad. It is unacceptable graphics on Microsoft Excel.

Response:

As pointed out, the graph created with Prism-GraphPad is inserted in the text.

#Point :

Discussion:

The authors should explorer the results in the second and third paragraphs. Because there is no information in both of them.

Response:

We are deeply grateful to you for your insights.

Based on comments from you and other reviewers,

We judged that the character of the second and third paragraphs of the discussion was more suitable for introduction. Because the two paragraphs consisted of contents that enhance the overall understanding of periodontitis (the keyword of this paper).

So the second and third paragraphs of Discussion that you mentioned were moved to Introduction and rewritten. (p. 2, line 50 – 70 , p. 2, line 92 - p. 3, line 107)

We would be grateful if you can review and comment on the changes.

Point:

Conclusion – The authors do not answer the aim of the study in the Conclusion. Why are you using the results of the survey in the Conclusion?

Please, rewrite the Conclusion, and answer the objective of the study.

Response:

We appreciate your valid and constructive comments.

In accordance with the title of the paper, the findings of this paper were briefly amended in the conclusion.

p.10, line 315 - 318

In this study, we found a statistically significant difference between the work patterns and prevalence of periodontitis in Korean women aged over 50 years. For female workers with irregular working patterns, the prevalence of periodontitis  was lower by 10.3% than that of workers with regular working patterns (40.3% vs 30.0%, p=0.011).

# Point:

References – Should be revised in the text.

Response :

The text has been revised overall. At the same time, references were added.

p.10, line 332

  1. Caton, J.G.; Armitage, G.; Berglundh, T.; Chapple, I.L.; Jepsen, S.; Kornman, K.S.; Mealey, B.L.; Papapanou, P.N.; Sanz, M.; Tonetti, M.S. A new classification scheme for periodontal and peri‐implant diseases and conditions–Introduction and key changes from the 1999 classification. Journal of periodontology 2018, 89, S1-S8.

Reviewer 2 Report

The purpose of this study was to examine the relationship of the prevalence of periodontitis with two classes of work schedules, regular and irregular, in the population over age 50 in Korea.  The study examined KNHANES data from the years 2013-2015.

In general, the study design is sound.  The comments below reflect specific and general observations about content and Discussion and Conclusions.

p. 4, line 120: In periodontal literature, the term "pocket" is used to measure the depth between the gingival crest and the depth of the sulcus (not "pouch")

The Results Section is not clear on differences between the regular and irregular work groups.  It also is confusing to indicate that one group was higher but not have statistically significant differences.  The only differences that should be reported are the ones with statistically significant differences, although there can be general references to no statistically significant differences between the two groups on the following variables: (give examples).  It would be helpful if there were more consistency in reporting the results and making consistent comparisons throughout the Section.

In Results, line 146-147 is an inaccurate statement.  For "most", there would need to be over 50%.  Perhaps the authors wish to restate the statement as the "mode" was elementary school or less.  This comment also applies to lines 147-148.  The authors need to be more specific about which group results they are stating and comparing.  

All of the Discussion on p.8 would be more appropriately included in the Introduction/Background Section.  The Discussion Section actually begins on p.9.  This Section would benefit from enhancing the analysis of the results.  For example, there is opportunity to discuss possible reasons for the increased prevalence of periodontal disease for women workers over 50 years of age with an irregular work schedule.  Additionally, line 271 states that the prevalence of periodontal disease was LOWER for women who work irregular shifts.  Is this consistent with the previous statement?  If not, please correct the narrative.  If it is consistent, please explain.  The discussion section is an opportunity for the authors to provide their INTERPRETATION of the results.

Although the authors refer to frequency of "annual health exam", the data table reflects "annual dental exam".  It would be more accurate to include the term annual dental exam in the body of the manuscript.

The conclusion of this study is not consistent with the title of the study.  The title indicates that the purpose of the study was to examine the relationship of periodontitis to work schedules; the conclusion seems to emphasize the importance of frequent oral exams.

Author Response

Response to Reviewer 2 Comments

Point :

p.4, line 120: In periodontal literature, the term "pocket" is used to measure the depth between the gingival crest and the depth of the sulcus (not "pouch")

Response:

Thank you for your pointing out. We’ve made it right.

p.4, line 158

Periodontal tissue status was assessed using the community periodontal index (CPI) scoring system (0, healthy; 1, bleeding; 2, presence of calculus; 3, periodontal pocket 4–5 mm; 4, periodontal pocket ≥6 mm).

#Point :

The Results Section is not clear on differences between the regular and irregular work groups. It also is confusing to indicate that one group was higher but not have statistically significant differences. The only differences that should be reported are the ones with statistically significant differences, although there can be general references to no statistically significant differences between the two groups on the following variables: (give examples). It would be helpful if there were more consistency in reporting the results and making consistent comparisons throughout the Section.

Response:

We appreciate your valid and constructive comments.

When we described the differences between the two groups, we also tried to describe the comparison between the two groups in more detail by mentioning the statistically insignificant differences.

As you pointed out, we were able to clarify the results section by deleting the description of the statistically insignificant differences.

Changes are reflected in the text and are marked in red. (the result section)

p.5, line 180

3.1. Characteristics of the study population at baseline

In total, 3320 participants were included in the study. Characteristics of the study population at baseline are shown in Table 1. The study population included 1779 men (53.6%) and 1543 women (46.4%) with a mean age of 60.86±7.32 years. In the Characteristics of the study population at baseline, we compared and analyzed family income, education level, smoking, alcohol, obesity, hypertension, diabetes and dyslipidemia.

There was a statistically significant difference in the educational level between the two study groups. (p<0.001).  In the irregular work pattern group, the percentage of "graduation from elementary school or lower" level " was highest. (37.1%). On the other hand, in the regular work pattern group, the percentage of "high school graduate" level was highest (36.9%).  There was a statistically significant difference in the smoking status and the proportion of patients with dyslipidemia between the two study groups. The proportion of smokers in the irregular work pattern group was higher than in the regular work pattern group (17.2% vs. 21.3; p = 0.014). The proportion of patients with dyslipidemia was high in the irregular work pattern group (19.9% vs 23.3%; p = 0.041).

There were no statistically significant differences in family income, alcohol, obesity including BMI, hypertension, and diabetes.

#Point:

In Results, line 146-147 is an inaccurate statement.  For "most", there would need to be over 50%.  Perhaps the authors wish to restate the statement as the "mode" was elementary school or less.  This comment also applies to lines 147-148.  The authors need to be more specific about which group results they are stating and comparing.

Response:

Thank you for your pointing out.

We used "MOST" in the most frequent sense, but as you pointed out, this seems inappropriate. The word "MOST" has been rewritten to convey the intended meaning without using the word. We’ve made it right.

p.5, line 186

There was a statistically significant difference in the educational level between the two study groups. (p<0.001).  In the irregular work pattern group, the percentage of "graduation from elementary school or lower" level " was highest. (37.1%). On the other hand, in the regular work pattern group, the percentage of "high school graduate" level was highest (36.9%).

#Point:

All of the Discussion on p.8 would be more appropriately included in the Introduction/Background Section. The Discussion Section actually begins on p.9. 

This Section would benefit from enhancing the analysis of the results. For example, there is opportunity to discuss possible reasons for the increased prevalence of periodontal disease for women workers over 50 years of age with an irregular work schedule. 

Response:

As you pointed out, part of content of the discussion (p8) was moved to the introduction and rewritten. You can check the changes in the introduction.

Thanks to your recommendation, this paper could have improved introduction and discussion.

#Point:

Additionally, line 271 states that the prevalence of periodontal disease was LOWER for women who work irregular shifts.  Is this consistent with the previous statement?  If not, please correct the narrative.  If it is consistent, please explain.  The discussion section is an opportunity for the authors to provide their INTERPRETATION of the results

Response: Thank you for your pointing out.

In women, the prevalence of periodontitis was significantly lower in the irregular work pattern group, and it is described in the results.

I reviewed carefully for any confusion and wrote it again, so I'd appreciate it if you check it out and point it out again.

# Point:

Although the authors refer to frequency of "annual health exam", the data table reflects "annual dental exam".  It would be more accurate to include the term annual dental exam in the body of the manuscript.

Response: We appreciate your keen comments.

In the case of “annual health examination”, which can be confirmed in Table 2, male and female were compared separately, and both were statistically significantly higher in irregular groups.

On the other hand, “annual dental examination”, which can be confirmed in Table 3, was statistically significantly higher in the irregular group only in women.

Since “the annual health examination” and “annual dental examination” identified in each table are tests of different personalities, the meaning of each was considered separately.

The reason why this paper mentions regular oral examinations is because it is considered as important as other risk factors for periodontitis.

# Point:

The conclusion of this study is not consistent with the title of the study. The title indicates that the purpose of the study was to examine the relationship of periodontitis to work schedules; the conclusion seems to emphasize the importance of frequent oral exams.

Response:

We appreciate your valid and constructive comments.

In accordance with the title of the paper, the findings of this paper were briefly amended in the conclusion.

p.10, line 315 - 318

In this study, we found a statistically significant difference between the work patterns and prevalence of periodontitis in Korean women aged over 50 years. For female workers with irregular working patterns, the prevalence of periodontitis  was lower by 10.3% than that of workers with regular working patterns (40.3% vs 30.0%, p=0.011).

Round 2

Reviewer 1 Report

The authors have addressed all issues raised by the reviewers.

Keep safe.

Author Response

June 2, 2020

Manuscript ID: ijerph-794869

Title: “Association of work patterns and periodontitis prevalence in Korean adults aged 50 years or older: A nationwide representative study.”

Response to Reviewer 1 Comments

The authors have addressed all issues raised by the reviewers.

Keep safe.

We thank you for your thoughtful suggestions and insights, which have enriched the manuscript and produced a more balanced and better account of the research.

We hope that the revised manuscript will better meet the requirements of the ‘International Journal of Environmental Research and Public Health’ for publication.

All of our authors wish you a safe stay.

Thank you.

Very sincerely yours,

Young Jin Ra, M.D.

Young Jin Ra, M.D.
Department of Family Medicine, Pusan National University School of Medicine,Yangsan, 50612, South Korea

Medical Research Institute, Pusan National University Hospital,Busan, 49241, South Korea

Reviewer 2 Report

The Reviewer thanks the authors for making manuscript modifications clearly identifiable in red.

There is no defined research question in the Introduction Section.  After giving all the information about periodontal disease and the systemic conditions associated with night shift workers, one would expect a research question such as, for example, "The purpose of this study was to investigate whether there were differences in periodontal health between regular workers and workers who worked night shifts.  In addition, we were interested in whether there were differences in periodontal status between those workers who worked regular shifts and those workers who worked irregular shifts."  It would be helpful to the reader to have a clearer understanding of what the authors were pursuing in undertaking this study.

Again, the conclusion cited at the end of the abstract does not relate to the title of the manuscript.  Either the title should be modified or the conclusion should relate to the prevalence of periodontitis in regular and irregular work patterns.  The Conclusions in lines 307-310 might be included in the Abstract Section, to replace the current Conclusion in the Abstract Section.

Lines 93-108 discuss conditions related to night workers.  Are the authors equating night shift workers with irregular shift workers?  If so, this needs to be explained.  If not, why is this section included? Are there inferences that can be made from the health behavior of night shift workers that are relevant to this study?

In the Discussion Section, lines 275-293 discuss the significant difference in periodontitis between women irregular shift workers (lower) and women regular shift workers.  However, there is more opportunity for the authors to reflect on what may contribute to these differences.  For example, the authors state that the irregular shift workers (women) had more frequent oral examinations than women regular shift workers.  Is it possible that the women irregular shift workers had more occasions to visit the dentist during the day (regular dental office hours?) than women regular shift workers, who would only have had occasion for regular (non-emergent) dental visits during evenings or weekends?  It is interesting that the women irregular shift workers had lower education levels than the women regular shift workers.  What are the implications of that for this study?  Do they have lower wages?  Would that mean that they do not buy as many "treat/sugar containing" foods (candy, cake, cookies, etc)?  Or to what do the author attribute the lower education level connection with lower prevalence of periodontitis in this group?

Since there were a large group of variables for which there were no significant differences, it would be helpful to summarize those in the Conclusions Section (beginning with line 307) to illustrate where there were no differences.

Author Response

June 2, 2020

Manuscript ID: ijerph-794869

Title: “Association of work patterns and periodontitis prevalence in Korean adults aged 50 years or older: A nationwide representative study.”

Response to Reviewer 2 Comments

#Point :

There is no defined research question in the Introduction Section. 

After giving all the information about periodontal disease and the systemic conditions associated with night shift workers, one would expect a research question such as, for example, "The purpose of this study was to investigate whether there were differences in periodontal health between regular workers and workers who worked night shifts.  In addition, we were interested in whether there were differences in periodontal status between those workers who worked regular shifts and those workers who worked irregular shifts."  It would be helpful to the reader to have a clearer understanding of what the authors were pursuing in undertaking this study.

Response:

We appreciate your valid and constructive comments.

We have added the research questions you recommended.

P.3 103 -105

The purpose of this study was to investigate whether there were differences in the prevalence of periodontitis between work patterns. In addition, we were interested in how important the “work patterns” plays in the difference in prevalence of periodontitis.

#Point :

Again, the conclusion cited at the end of the abstract does not relate to the title of the manuscript.  Either the title should be modified or the conclusion should relate to the prevalence of periodontitis in regular and irregular work patterns. 

The Conclusions in lines 307-310 might be included in the Abstract Section, to replace the current Conclusion in the Abstract Section.

Response:

Thank you for your pointing out. We changed as you recommended.

P.1 30-32

In conclusion we found a statistically significant difference between the work patterns and prevalence of periodontitis in Korean women aged over 50 years.

#Point :

Lines 93-108 discuss conditions related to night workers.  Are the authors equating night shift workers with irregular shift workers?  If so, this needs to be explained.  If not, why is this section included? Are there inferences that can be made from the health behavior of night shift workers that are relevant to this study?

Response:

We appreciate your keen comments.

The authors conducted a study divided into two groups by work type to investigate the difference in the prevalence of periodontitis according to the work pattern.

Based on the self-report questionnaire question, it is divided into one group that only works normal 8-hour day (9:00 am to 5:00 pm) and one that works irregularly (outside the normal 8-hour day). So night shift workers are included in irregular work groups.

I reviewed carefully for any confusion and wrote it again, so I'd appreciate it if you check it out and point it out again.

P.4 140 - 145

2.3. Work pattern

According to their responses to the questionnaire, the study participants were categorized according to the regular or irregular work pattern group. Regular work pattern was defined as work performed the normal 8-hour day (9:00 am to 5:00 pm). Irregular work pattern was defined as work performed outside the normal 8-hour day (9:00 am to 5:00 pm) and included early morning, late afternoon, night work including any kind of shift work.

#Point :

In the Discussion Section, lines 275-293 discuss the significant difference in periodontitis between women irregular shift workers (lower) and women regular shift workers.  However, there is more opportunity for the authors to reflect on what may contribute to these differences.  For example, the authors state that the irregular shift workers (women) had more frequent oral examinations than women regular shift workers.  Is it possible that the women irregular shift workers had more occasions to visit the dentist during the day (regular dental office hours?) than women regular shift workers, who would only have had occasion for regular (non-emergent) dental visits during evenings or weekends? 

Response :

We appreciate your valid and constructive comments.

We have added more explanation as you recommended.

P.9 284 – 287

This means that workers with irregular working patterns were given more opportunities to receive oral examinations more frequently than those who had to work during the daytime (9:00 am to 5:00 pm) and this contributes to the prevalence of periodontitis for female workers.

#Point :

It is interesting that the women irregular shift workers had lower education levels than the women regular shift workers. What are the implications of that for this study?  Do they have lower wages?  Would that mean that they do not buy as many "treat/sugar containing" foods (candy, cake, cookies, etc)?  Or to what do the author attribute the lower education level connection with lower prevalence of periodontitis in this group?

Response :

We appreciate your keen comments.

Overall, Korea's industrial structure is passing through a turbulent period of mixed regular and non-regular workers.

The regular workers here are usually groups that are guaranteed fixed day work (9-17).

For non-regular workers, include groups that have short or irregular shifts or are not guaranteed regular work hours.

Overall, full-time employees with good working conditions have higher average education and therefore higher wages.

On the other hand, non-regular workers have a lower average education and therefore generally have poor working conditions and worse wages and welfare benefits.

If the overall wage is low, you'll be more likely to eat unhealthy foods (e.g., instant foods) that are cheaper than high-quality foods, which are ultimately linked to periodontal disease.

This paper does not describe the above arguments because it is not a topic for nutrition and dental disease.

The authors of this paper wanted to study the causal relationship between these working conditions and various chronic diseases. I feel sorry for the above reality that a lot of data shows in the course of the research.

Thanks to your point, I am very grateful to be able to look at the present reality once again and get hints on future research topics.

#Point :

Since there were a large group of variables for which there were no significant differences, it would be helpful to summarize those in the Conclusions Section (beginning with line 307) to illustrate where there were no differences.

Response:

We appreciate your valid and constructive comments.We have added the research results explanation you recommended.

P.10 305 – 307

In contrast, there was no statistically significant difference in the prevalence of the periodontitis in male workers over 50 years of age according to their work patterns (55.1% vs 56.40%, p=0.742).

We thank you for your thoughtful suggestions and insights, which have enriched the manuscript and produced a more balanced and better account of the research.

We hope that the revised manuscript will better meet the requirements of the ‘International Journal of Environmental Research and Public Health’ for publication.

All our authors look forward to your positive reply.

Very sincerely yours,

Young Jin Ra, M.D.
Department of Family Medicine, Pusan National University School of Medicine,

Yangsan, 50612, South Korea

Medical Research Institute, Pusan National University Hospital,

Busan, 49241, South Korea
